# Not the Cryptic Species: Diversity of *Hipposideros gentilis* (Chiroptera: Hipposideridae) in Indochina

**Alexander P. Yuzefovich [1], Ilya V. Artyushin [1,\*] and Sergei V. Kruskop [2,\*]**

[1] Department of Vertebrate Zoology, Moscow State University, Leninskiye Gory 1 bld. 12, 119234 Moscow, Russia; yuzefovich2015elf@gmail.com

[2] Zoological Museum, Moscow State University, Bolshaya Nikitskaya 2, 125009 Moscow, Russia

\* Correspondence: sometyx@gmail.com (I.V.A.); kruskop@zmmu.msu.ru (S.V.K)

**Abstract:** We present here the result of phylogenetic analysis for Vietnamese *Hipposideros gentilis* specimens using 7 nuclear genes and one mitochondrial gene. The complex distribution of divergent mitochondrial DNA lineages contradicts, at least in part, nuclear and morphological data. The most likely explanation for this discordance is the historical hybridization between ancestral populations of *H. gentilis* and *H. rotalis*/*H. khaokhouayensis*. Our data supports the species status of *H. gentilis*, while only partially corroborating its previously proposed subspecies delimitation. We suggest the lowland forest populations from south Vietnam may correspond to their own subspecies. At the same time, the close phylogenetic relationship and morphological similarity of mountain forms from south and central Vietnam to the north Vietnamese populations make doubtful the subspecies status of *H. gentilis sinensis*.

**Keywords:** roundleaf bats; *Hipposideros*; cryptic diversity; Southeast Asia; nuclear genes

## 1. Introduction

*Hipposideros* is the core genus to Hipposideridae (Old World leaf-nosed bats), a bat taxon prominent in Southeast Asia. It is also one of the largest mammalian genera, retaining a highly complicated taxonomy even after having lost two of its former species groups, namely "commersoni" and "cyclops" [1,2]. Within the genus, *Hipposideros bicolor* is a group that comprises the most forms. Many of them are similar in morphology, and their taxonomic placement is often uncertain. In the last decade, significant efforts were dedicated to solving this puzzle (e.g., [3–5]).

One of the common Southeast Asian species of leaf-nosed bats from the "bicolor" group was widely referred to as *Hipposideros pomona* by Andersen in 1918 [6–8]. Morphologically, this form was once treated as a subspecies or group of subspecies within *H. bicolor* (Temminck, 1834) [9]. Thereafter *H. pomona* was considered a distinct species, including the forms *gentilis* (Andersen, 1918) and *sinensis* (Andersen, 1918) as partial synonyms [10]. *Paracoelops megalotis* (Dorst, 1947) was initially described as a separate species and genus from central Vietnam [11]. However, a review of the single type specimen concluded that it belongs to *H. pomona* [12]. Molecular studies have revealed that several geographically limited haplogroups are present in *H. pomona* populations. Distances between these haplogroups are high enough to assume species-level divergence. Moreover, the monophyly of *H. pomona* was questioned [13,14]. In the Indochina peninsula, four main lineages have been discovered, namely northern, central, and two in the south. One of the southern clades takes a sister position to the lineage which includes *H. khaokhouayensis* (Guillen-Servent, Francis, 2006) and *H. rotalis* (Francis, Kock, Habersetzer, 1999) [13]. Some of these haplogroups were treated as subspecies in an earlier study of *H. pomona* genetic variability in China [5]. The clades SM and SY were assigned to the subspecies *sinensis* and *gentilis*, respectively, with no suggestions for the "central" clade (SH). The southern haplogroups were not discussed in that paper.

On the other hand, the shape of baculum (*os penis*) is very similar in different genetic lineages of *H. pomona* [15] (see also [16]). The baculum is greatly reduced and stick-like, about 0.5 mm in length. It is strikingly different from that of *H. bicolor*, *H. cineraceus* (Blyth, 1853) and other related species [4]. At the same time *H. pomona* s. str. from southwest India is remarkably distinct from all other populations in bacular and cranial morphology [17]. *Hipposideros gentilis* is the next senior name, and in this case should be accepted as valid for all populations under current study.

Similarity in baculum shape, as well as the absence of obvious differences between populations in the skull morphometry and the structure of the nasal leaves (orig., unpubl.), cast a shadow of doubt upon the putative species status of the *H. gentilis* haplogroups. Here, we apply a multi-locus analysis of nuclear genes to address this subject.

## 2. Materials and Methods

### 2.1. Analyzed Specimens

Our research is based on the Zoological Museum of Moscow State University (ZMMU) collections. Tissue samples (muscle fragments) were taken from specimens preserved in ethanol. In total, 14 specimens of *H. gentilis* (*H. "pomona"*) were used, together with a few other species of the genus Hipposideros. *Aselliscus stoliczkanus* was taken as outgroup. A full list of specimens and obtained sequences is provided in Tables 1 and 2. Also, additional sequences of *cytb* and four nuclear genes were downloaded from GenBank (see Table 2 and Appendix A).

**Table 1.** List of specimens from ZMMU used in the *cytb* analysis.

| Mus. ID | Species | Locality | | Accession |
|---------|---------|----------|---|-----------|
| S-190280 | *A. stoliczkanus* | N. Vietnam | Cat Ba | MZ219226 |
| S-167159 | *H.* cf. *armiger* | C. Vietnam | Quan Binh | MZ219210 |
| S-195483 | *H.* cf. *griffini* | S. Vietnam | Dong Nai | MZ219216 |
| S-186724 | *H. cineraceus* | S. Vietnam | Con Dao | MZ219220 |
| S-186725 | *H. cineraceus* | S. Vietnam | Con Dao | MZ219221 |
| S-186730 | *H. cineraceus* | S. Vietnam | Con Dao | MZ219222 |
| S-191867 | *H. cineraceus* | S. Vietnam | Dong Nai | MZ219218 |
| S-195484 | *H. cineraceus* | S. Vietnam | Dong Nai | MZ219219 |
| S-186567 | *H. diadema* | S. Vietnam | Binh Phuoc | MZ219211 |
| S-191868 | *H. galeritus* | S. Vietnam | Dong Nai | MZ219225 |
| S-167170 | *H. gentilis* | C. Vietnam | Quan Binh | MZ219215 |
| S-167171 | *H. gentilis* | C. Vietnam | Quan Binh | MZ219214 |
| S-167172 | *H. gentilis* | C. Vietnam | Quan Binh | MZ219223 |
| S-167173 | *H. gentilis* | C. Vietnam | Quan Binh | MZ219224 |
| S-190298 | *H. gentilis* | N. Vietnam | Cat Ba | MZ219212 |
| S-190301 | *H. gentilis* | N. Vietnam | Cat Ba | MZ219213 |
| S-190302 | *H. gentilis* | S. Vietnam | Bihn Chau | MZ219227 |
| S-191870 | *H. gentilis* | S. Vietnam | Dong Nai | MZ219217 |
| S-198154 | *H. gentilis* | SC. Vietnam | Gia Lai | MZ219228 |
| S-189221 | *H. grandis* | S. Vietnam | Dong Nai | MZ219209 |

**Table 2.** List of specimens used in the nuclear genes analysis.

| species | ID No | Reference | Locality | THY | ABHD | ROGDI | RAG2 | ACOX | COPS | SORBS |
|---|---|---|---|---|---|---|---|---|---|---|
| *H. gentilis* | S-167174 | original | C Vietnam | | MZ219110 | | | MZ219140 | | MZ219182 |
| *H. gentilis* | S-190298 | original | N. Vietnam | MZ219096 | | | | MZ219134 | | |
| *H. gentilis* | S-190299 | original | N. Vietnam | MZ219097 | | | | MZ219135 | | |
| *H. gentilis* | S-190301 | original | N. Vietnam | | MZ219118 | MZ219199 | | MZ219133 | | MZ219179 |
| *H. gentilis* | S-190302 | original | S Vietnam | MZ219098 | MZ219119 | MZ219200 | MZ219170 | MZ219137 | MZ219160 | MZ219180 |
| *H. gentilis* | S-191870 | original | S Vietnam | MZ219099 | MZ219122 | MZ219203 | MZ219172 | MZ219138 | MZ219159 | MZ219181 |
| *H. gentilis* | S-191871 | original | S Vietnam | | MZ219123 | MZ219204 | | MZ219139 | | |
| *H. gentilis* | S-191909 | original | S Vietnam | | MZ219124 | MZ219205 | MZ219173 | MZ219136 | | |
| *H. gentilis* | S-198253 | original | SC Vietnam | MZ219107 | | | MZ219175 | MZ219152 | | MZ219191 |
| *H. gentilis* | S-198254 | original | SC Vietnam | MZ219108 | MZ219129 | MZ219207 | MZ219176 | MZ219153 | | MZ219192 |
| *H. gentilis* | S-198255 | original | SC Vietnam | MZ219109 | | MZ219208 | MZ219177 | MZ219154 | | MZ219193 |
| *H. abae* | ML162-210211-HIPABA | [1] | Mali | KP176357 | KP176214 | KP176320 | KP176020 | | | |
| *H. armiger* | S-195483 | original | S Vietnam | MZ219094 | MZ219127 | MZ219206 | MZ219174 | MZ219132 | MZ219157 | MZ219190 |
| *H. armiger* | T-171109-1 | [1] | Vietnam | KP176354 | KP176210 | KP176317 | KP176016 | | | |
| *H. centralis* | S-192897 | original | Ethiopia | MZ219105 | MZ219125 | | | MZ219149 | | |
| *H. cf. abae* | S-189528 | original | Ethiopia | MZ219104 | MZ219116 | | | MZ219150 | | |
| *H. cf. grandis* | S-195421 | original | SC Vietnam | MZ219093 | MZ219126 | | | MZ219131 | MZ219156 | MZ219189 |
| *H. cf. ruber* | ML29-310111-HIPCAF | [1] | Mali | KP176358 | KP176215 | KP176321 | KP176021 | | | |
| *H. cineraceus* | S-167179 | original | C Vietnam | | MZ219111 | | | MZ219146 | | |
| *H. cineraceus* | S-186724 | original | S Vietnam | MZ219100 | | | MZ219165 | MZ219143 | MZ219161 | MZ219185 |
| *H. cineraceus* | S-186725 | original | S Vietnam | MZ219101 | MZ219113 | | MZ219166 | MZ219144 | | MZ219186 |
| *H. cineraceus* | S-186730 | original | S Vietnam | MZ219102 | MZ219114 | MZ219195 | MZ219167 | MZ219145 | | |
| *H. cineraceus* | S-191867 | original | S Vietnam | MZ219103 | MZ219120 | MZ219201 | MZ219171 | MZ219147 | MZ219162 | MZ219183 |
| *H. cineraceus* | S-195484 | original | S Vietnam | | MZ219128 | | | MZ219142 | | MZ219184 |
| *H. diadema* | 10.0011 | [1] | Thailand | KY552688 | KY552683 | KY552686 | | | | |
| *H. diadema* | S-186567 | original | S. Vietnam | MZ219095 | MZ219112 | MZ219194 | MZ219164 | | MZ219158 | |
| *H. galeritus* | S-191868 | original | S. Vietnam | | MZ219121 | MZ219202 | | MZ219148 | | MZ219187 |
| *H. galeritus* | T-090708-6 | [1] | Vietnam | KP176355 | KP176211 | | KP176017 | | | |

**Table 2.** *Cont.*

| species | ID No | Reference | Locality | THY | ABHD | ROGDI | RAG2 | ACOX | COPS | SORBS |
|---|---|---|---|---|---|---|---|---|---|---|
| *H. grandis* | S-189221 | original | S Vietnam | MZ219092 | MZ219115 | MZ219196 | MZ219168 | MZ219130 | MZ219155 | MZ219178 |
| *H. halophyllus* | Hhal2 | [1] | Thailand | KP176359 | KP176216 | KP176322 | KP176022 | | | |
| *H. jonesi* | ML155BIS-160211-HIPJON | [1] | Mali | | KP176213 | KP176319 | KP176019 | | | |
| *H. khaokhouayensis* | S-190294 | original | N Vietnam | | MZ219117 | MZ219198 | | MZ219141 | MZ219163 | |
| *H. khaokhouayensis* | T-070108-1 | [1] | Vietnam | KY552735 | KY552730 | KY552733 | | | | |
| *H. larvatus* | Hlar29 | [1] | Thailand | KP176360 | KP176217 | KP176323 | KP176023 | | | |
| *H. "ater"** | T-250608-2 | [1] | Vietnam | KY552740 | KY552736 | | | | | |
| *H. "pomona"** | T-180809-3 | [1] | Vietnam | KP176356 | KP176212 | KP176318 | KP176018 | | | |
| *A. stoliczkanus* | S-190280 | original | N Vietnam | MZ219106 | | MZ219197 | MZ219169 | MZ219151 | | MZ219188 |

* Names of sequences taken from Genbank follow those used by the original authors.

*2.2. Morphometric Analysis.*

To understand how the observed genetic diversity is reflected in morphology, we performed a morphometric analysis using cranial and dental measurements.

A total of 102 specimens of the *Hipposideros pomona/gentilis* complex (dry or alcohol preserved skins with extracted skulls) were examined for morphometric comparison. We excluded several specimens due to missing measurements. Only 99 were included in the final analysis. The full list of these specimens is provided in Appendix C. Localities are shown in Figure 1. Acronyms of the processed collection repositories are: MNH—British Museum of Natural History (London, UK); MNHN—National Museum of Natural History (Paris, France); ROM—Royal Ontario Museum (Toronto, Canada); NHW—Vienna Museum of Natural History (Vienna, Austria); ZMMU—Zoological Museum of Moscow State University (Moscow, Russia); ZMB—Berlin Zoological Museum (Berlin, Germany).

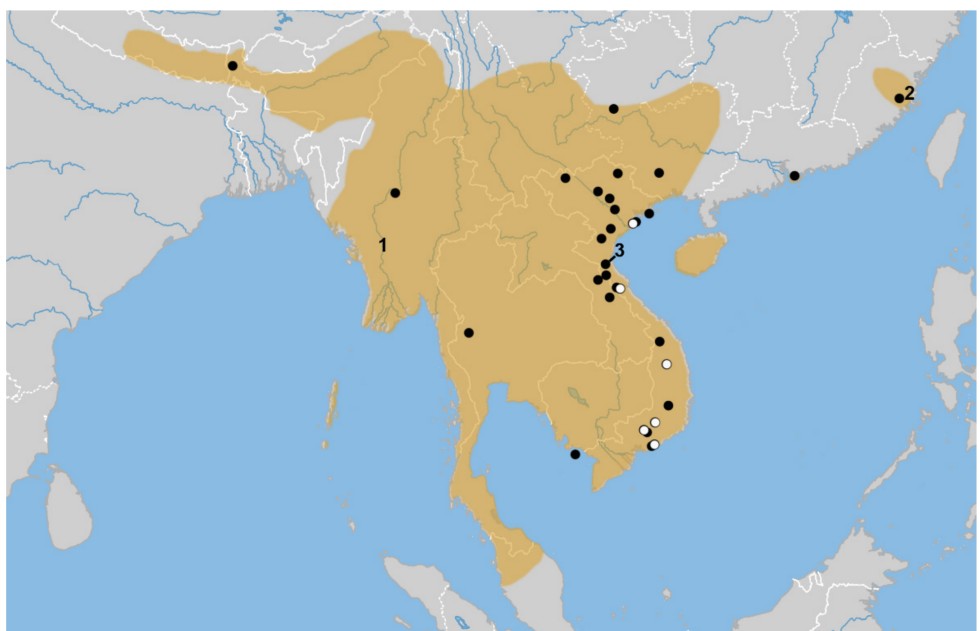

**Figure 1.** Distribution range of *Hipposideros gentilis* (adopted from [18]). Localities of morphologically studied material are marked with full circles; localities for original genetic samples (*cyt*b and nuclear genes) are marked with empty circles. Type localities of named forms are numbered as follows: 1—*gentilis* Andersen, 2—*sinensis* Andersen, 3—*megalotis* Dorst.

Cranial and dental measurements were made under stereo microscope with digital calipers, rounding to the nearest 0.01 mm. The following cranial measurements were taken: greatest skull length (TL), condylo-canine length (CCL), skull width at the mastoid (MW), brain case width above the mastoids (BCW), occiput height (height from the lower margins of the occipital condyles to the highest point just above them; OH), zygomatic width (ZW), width of the postorbital constriction (POC), rostral width at the level of the anterorbital foramina (RW), length of rostrum in front of the anterorbital foramen (RL), width across the upper canines (CC), width across posterior upper molars (MM), length of the upper tooth row (CM), length of the upper molariform row, distance from $P^4$ to the posterior molar (PM), longitudinal length of the upper canine (C), width of nasal opening (NO), lower tooth row length (cm), articular length of the mandible (MdL), and height of the mandible (MdH).

To assess the variation pattern of quantitative characters, Principal Component (PC) and Discriminant Function (DF) analyses were performed for the 20 craniodental measurements, using STATISTICA for Windows version 9.0 (StatSoft, currently TIBCO Software Inc., Palo Alto, CA, USA). DF analysis was used to calculate squared Mahalanobis distances between groups and the significance of inter-group differences. The training set for

calculating the squared Mahalanobis distances for the DF analysis included six samples, namely "Myanmar", "Central Thailand", "NW Indochina", "NE Indochina and SE China", "C Indochina" and "S Indochina". The third and fourth training sets were selected based roughly on the two haplogroup (SY and SM) ranges as they are described in [5]. Indian material, due to low specimens' availability, was not treated as a training set, but was included in the analysis as "unidentified".

We were not able to genotype all specimens used in the morphometric study. However, all geographic samples from Indochina used in both the PS and DF analyzes include specimens genotyped for one or both mitochondrial genes and for nuclear genes (see Appendix C and appropriate Figures).

### 2.3. Molecular Analysis

Genomic DNA was extracted from ethanol-preserved muscle tissue by standard method of phenol–chloroform deproteinization [19]. One mitochondrial gene cytochrome b (*cytb*; 1137 bp) and fragments of seven nuclear genes (ABHD11, 460 bp; ACOX2, 597 bp; COPS, 744 bp; RAG2, 1035 bp; ROGDI2, 509 bp; SORBS2, 569 bp; and THY, 565 bp) were sequenced. Primers were taken from previously published papers [20–23]. The thermal profile amplification reaction included 35 cycles and was set up as follows: denaturation for 30 s at 94 °C, annealing for 1 min (at 55 °C for *cytb* and 60°C for the nuclear genes), and elongation for 1 min at 72 °C. Predenaturation lasted 3 min at 94 °C, and the final elongation was 6 min at 72 °C. Automatic sequencing was performed on an ABI PRISM 3500xl sequencer (Applied Biosystems (ABI), Thermo Fisher Scientific, Waltham, Massachusetts, United States) using ABI PRISM® BigDyeTM Terminator v. 3.1 reagent kits (Applied Biosystems, United States) in the laboratory of Eurogen Joint Stok Company (Moscow, Russia). To obtain additional mitochondrial topology, a total of 117 specimens, including 95 sequences of *H. gentilis* and seven other Asian *Hipposideros* species were analyzed for patterns of genetic divergence in the COI region. These data were taken from publicly accessible projects housed by the Barcode of Life Data System (BOLD, Guelph, Ontario, Canada) [24]. A list of BOLD Process ID numbers is provided in Appendix B.

Obtained sequences were aligned by the MAFFT v7.307 algorithm [25], then trimmed and corrected manually, after which heterozygous sites were marked by IUPAC ambiguity symbols for tree construction and phased in DNA 6.12.03 [26] for network building. Indels were replaced by "N." The reconstruction of phylogenetic trees was carried out using maximum likelihood (ML) in IQ-TREE v2.0.6 [27,28]. The built-in algorithm was applied to select optimal models and partitioning scheme [29]. Bayesian analysis (BA) was also performed using the MrBayes 3.2.6 program [30–32]. Partitioning schemes and models for BA were determined in Partition-Finder v. 2.1.1 [33] using a BIC criterion with a greedy search method. The initial partition scheme included separate partitions for each gene. For mitochondrial genes, codon positions were used as the initial partitioning scheme.

Supertree was computed using the MRP method (matrix representation with parsimony) [34–36]. Using the alignment of individual nuclear genes, bootstrap samples of 1000 ML trees were constructed. The resulting topologies were combined using the maximum parsimony method, implemented in the *phytools* v0.7.70 [37] package in the *R* v 3.5.2 [38] environment. The reliability of the topology was calculated by applying a bootstrap procedure to total tree sample.

## 3. Results

### 3.1. Pattern of Variation in the mtDNA Marker

Phylogeny reconstructed from *cytb* sequences mostly corroborates previously published COI gene data [13]. Basal nodes and relationships between different species groups have low support. The monophyly of the species groups generally is well supported. Topologies of Bayes and ML trees didn't contradict each other at highly supported nodes (Figure 2).

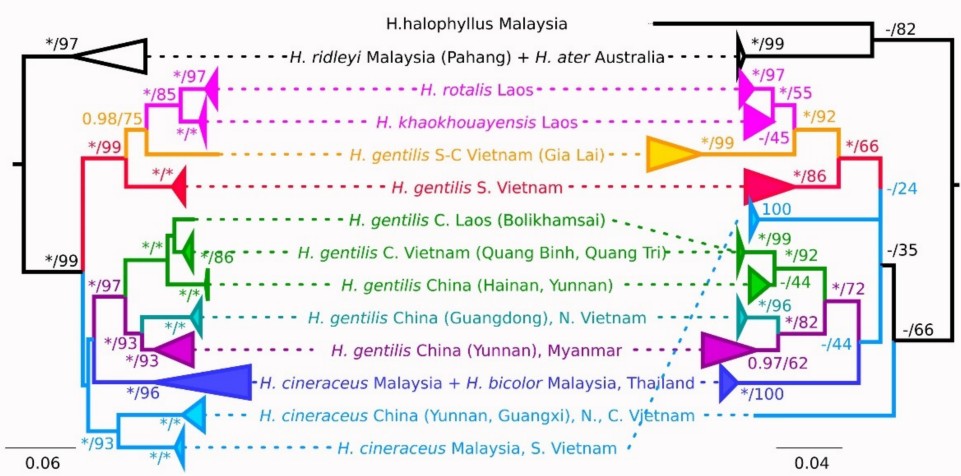

**Figure 2.** Phylogenetic relationships of selected *Hipposideros* species inferred from *cytb* (left) and COI (right) sequences. Topologies for both genes are constructed using maximum likelihood (ML). Bayesian posterior probabilities/maximum likelihood bootstrap supports are shown at the tree nodes. Only bootstrap support values over 70 are shown; maximum supports (Bayesian probability 0.99 and higher, 100% ML) are denoted by asterisks. Dashes denote nodes, which didn't appear on MB tree. Scale bars indicate genetic distances estimated by the ML method. The bootstrap values are derived from 1000 replications.

*Hipposideros gentilis* is non-monophyletic by this gene and is divided into two well-supported clades (Figure 2, Figure S1; Table 3, Table S1). The first clade is by itself para-phyletic with respect to the *H. khaokhouayensis/rotalis* lineage and includes samples from the southern lowland forests of Vietnam (Dong Nai, Tuy Phong, and Bihn Chau provinces) and from the southern part of the Vietnamese central highlands (Gia Lai province). Within the second clade of *H. gentilis*, specimens from Vietnam, Laos and China are represented. The Haplogroup from the northern half of central Vietnam forms a sister clade to the animals from Laos, Hainan (SH clade in [5]) and one specimen from Yunnan province, China. Animals from north Vietnam (Gulf of Tonkin) are grouped with those from Guangdong (SM clade). Such relations somewhat contradict the results of [5] and suggest a possible overlap in distribution of haplogroups. Both the central Vietnam and Gulf of Tonkin clades are monophyletic with high support.

### 3.2. Pattern of Variation in nuDNA Markers

Analyses of individual nuclear genes mainly demonstrate insufficient phylogenetic signal (Figure S2, Tables S2, S3). Three of seven genes (RAG2, ROGDI, THY) yielded monophyletic clades for the analyzed samples of *H. gentilis*. However, only the THY gene had a good bootstrap support value for this branch (87%). It is unclear which of the two species (*H. ater*, *H. cineraceus*) is closest to *H. gentilis*. The other three genes (ABHD11, ACOX2, SORBS2) did not support *H. gentilis* monophyly, and we placed *H. khaokhouayensis*, *H. bicolor* or *H. cineraceus* respectively within this clade. Not enough sequences were obtained for the COPS gene to draw any meaningful conclusions.

Low diversity levels and dataset incompleteness precluded reliable distance computation, although the substitution count between geographic samples of *H. gentilis* is generally lower than those between species in the "bicolor" group. We also found heterozygous SNPs in sites which bear differences between the three Vietnamese *H. gentilis* lineages.

These data definitely contradict to the results obtained from mitochondrial genes and generally indicate closer relationships of *H. gentilis* populations.

**Table 3.** Between-group pairwise p-distances for *H. gentilis* and related forms. Distances are shown below the diagonal, and standard error estimates are shown above the diagonal. Groups correspond to the *cytb* tree branches on Figure 2.

| | *H. rotalis* | *H. khaokhouayensis* | *H. cineraceus* C. Vietnam | *H. cineraceus* S. Vietnam | *H. cineraceus* Malaysia | *H. gentilis* S. Vietnam | *H. gentilis* S-C. Vietnam | *H. gentilis* C. Vietnam | *H. gentilis* N. Vietnam | *H. gentilis* Yunnan | *H. gentilis* Hainan | *H. gentilis* Laos |
|---|---|---|---|---|---|---|---|---|---|---|---|---|
| *H. rotalis* | | 0.0054 | 0.0084 | 0.0093 | 0.0080 | 0.0072 | 0.0082 | 0.0086 | 0.0088 | 0.0082 | 0.0086 | 0.0085 |
| *H. kha* | 0.0402 | | 0.0087 | 0.0094 | 0.0081 | 0.0081 | 0.0087 | 0.0090 | 0.0088 | 0.0082 | 0.0091 | 0.0089 |
| *H. cin* C. Vietnam | 0.1021 | 0.0968 | | 0.0076 | 0.0071 | 0.0088 | 0.0093 | 0.0073 | 0.0083 | 0.0073 | 0.0079 | 0.0073 |
| *H. cin* S. Vietnam | 0.0988 | 0.0977 | 0.0803 | | 0.0076 | 0.0091 | 0.0094 | 0.0089 | 0.0087 | 0.0077 | 0.0086 | 0.0087 |
| *H. cin* Malaysia | 0.1092 | 0.1068 | 0.0965 | 0.0989 | | 0.0079 | 0.0087 | 0.0075 | 0.0079 | 0.0070 | 0.0080 | 0.0080 |
| *H. gen* S. Vietnam | 0.0753 | 0.0771 | 0.0953 | 0.0991 | 0.1028 | | 0.0095 | 0.0089 | 0.0088 | 0.0080 | 0.0089 | 0.0091 |
| *H. gen* S-C. Vietnam | 0.0683 | 0.0726 | 0.0985 | 0.0964 | 0.1037 | 0.0874 | | 0.0092 | 0.0093 | 0.0089 | 0.0092 | 0.0094 |
| *H. gen* C. Vietnam | 0.1007 | 0.0960 | 0.0886 | 0.0933 | 0.0988 | 0.0957 | 0.0985 | | 0.0076 | 0.0065 | 0.0056 | 0.0047 |
| *H. gen* N. Vietnam | 0.0980 | 0.0961 | 0.0904 | 0.0944 | 0.1018 | 0.0921 | 0.0964 | 0.0739 | | 0.0061 | 0.0082 | 0.0079 |
| *H. gen* Yunnan | 0.0994 | 0.0949 | 0.0896 | 0.0874 | 0.0981 | 0.0933 | 0.0934 | 0.0663 | 0.0583 | | 0.0069 | 0.0069 |
| *H. gen* Hainan | 0.1039 | 0.0989 | 0.0891 | 0.0911 | 0.1033 | 0.0977 | 0.0951 | 0.0417 | 0.0781 | 0.0769 | | 0.0058 |
| *H. gen* Laos | 0.0976 | 0.0937 | 0.0862 | 0.0904 | 0.1040 | 0.0973 | 0.1020 | 0.0266 | 0.0797 | 0.0722 | 0.0412 | |

However, the tree built with the MRP algorithm on the combined data for all the seven nuclear genes supports the monophyly of *H. gentilis* and provides well-resolved topology within its clade (Figure 3). The position of this clade relative to *H. bicolor, H. ater* and *H. khaokhouayensis* is unresolved, and *H. cineraceus* is a sister branch to all of the above. Contrary to mitochondrial data, animals from the south Vietnam lowlands form their own branch, which is the most basal within the *H. gentilis* radiation.

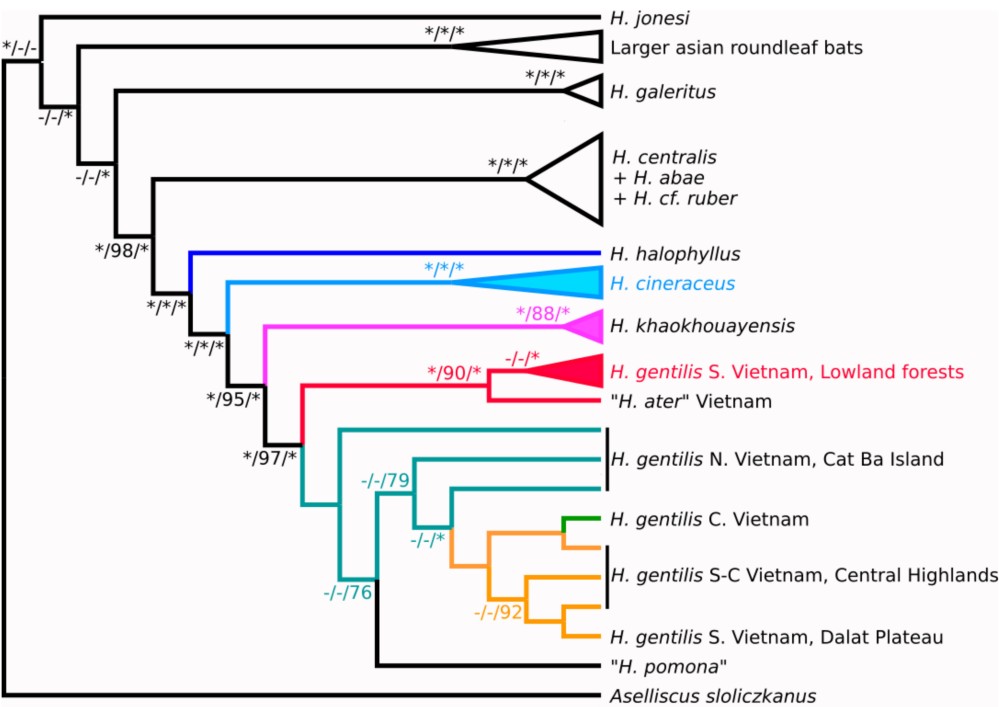

**Figure 3.** Phylogenetic tree based on seven nuclear gene topologies combined by MRP method. Bayesian/maximum likelihood/MRP supports are shown at the tree nodes. Maximum supports are denoted with asterisks. Dash symbols denote nodes, which didn't appear on respective tree.

Specimens from the Vietnamese central highlands are separated from the south Vietnam lowland populations and tend to aggregate with the specimens from other mountainous populations from the Dalat Plateau and central Vietnam. Animals from the north-east of Vietnam (Gulf of Tonkin) are paraphyletic with respect to the mountain populations.

### 3.3. Patterns of Morphological Variation

The morphometric data are moderately factorized: the total contribution of the first four factors to the total variance is about 70%, which means relatively low overall variation of cranial proportions in *H. pomona* s. lato. In the space of the first two factors, most geographic samples overlap highly, except for the Indian specimens, representing *H. pomona* s.str. (Figure 4). This agrees with the species-level dissimilarity between *H. pomona* and *H. gentilis*. The third factor greatly reduces the overlap between the samples from northern Indochina (both western and eastern) and the other analyzed samples. Then, the fourth factor separates the sample from southern Indochina from all the others (Figure 5). The most significant contribution to the first PC was made by CM, PM, C and cm variables (i.e., by the tooth row length); by MW, BCW, and CCL (skull width and overall size) to the second; and by CC and, to a lesser extent, other measurements related to skull width and mandible size (Table 4) to the fourth.

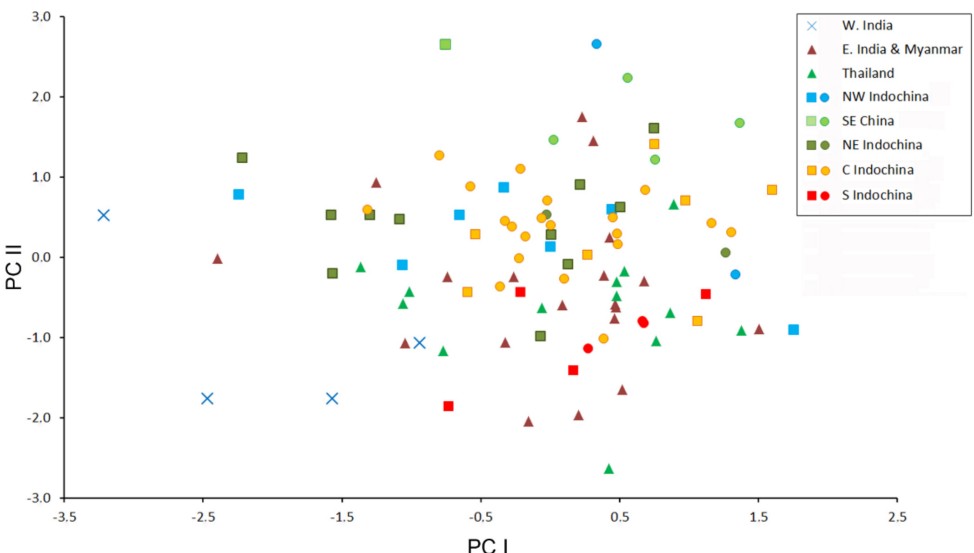

**Figure 4.** Bivariate scatter plot for the first two factors of a Principal Component analysis based on 18 cranial and dental measurements of 99 specimens of *Hipposideros pomona* s. lato. Specimens marked by closed circles were genotyped with at least one mitochondrial gene.

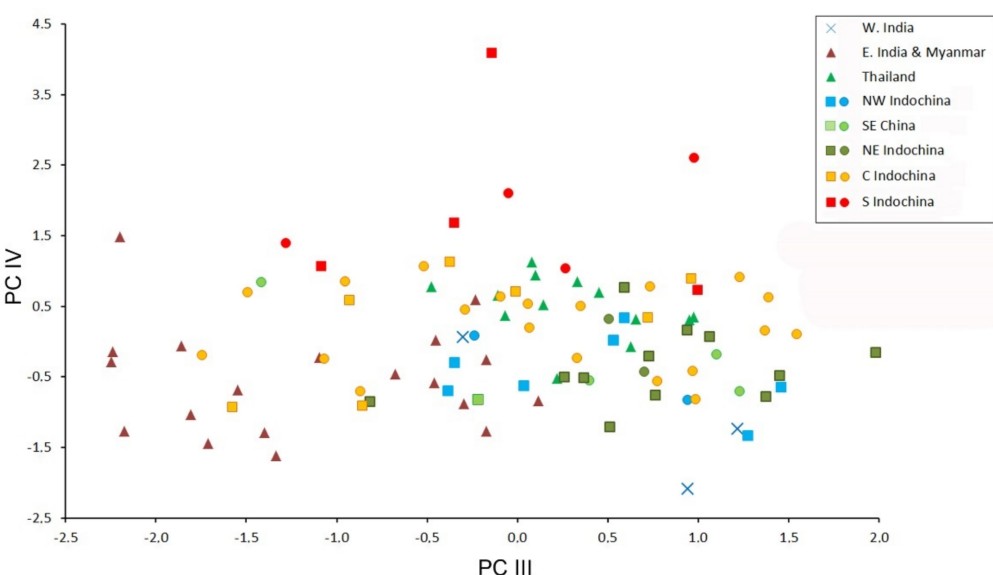

**Figure 5.** Bivariate scatter plot for the third and fourth factors of a Principal Component analysis based on 18 cranial and dental measurements of 99 specimens of *Hipposideros pomona* s. lato. Specimens marked by closed circles were genotyped with at least one mitochondrial gene.

The results of the discriminant function analysis clearly show the high significance of the differences among all the training sets, with one exception: animals from western and eastern north Indochina are separated insignificantly, suggesting their taxonomic equality (Figure 6; Table 5). Surprisingly, the specimens set from Myanmar turned out to be one of the most distinct. The training set from southern Indochina is well-separated from all the rest by the values of the first two canonical variables. It shows some degree of similarity with the set from central Thailand and overlaps slightly with it, but not the others. The specimens from the Dalat Plateau and Gia Lai province, which are similar in mitochondrial sequences to the south Vietnamese ones, show similarities with different training sets (more often with central Indochina), but not with southern Indochina.

**Table 4.** Results of PCA applied to specimens of the *Hipposideros pomona/gentilis* species complex based on 18 cranial measurements: factor loadings, eigenvalues, and percentage of total variance. See text for measurement abbreviations. Factor loadings over 0.7 are highlighted in boldface type.

| | PC I | PC II | PC III | PC IV |
|---|---|---|---|---|
| TL | 0.3874 | 0.5700 | 0.2348 | 0.5591 |
| CCL | 0.4756 | 0.6068 | 0.1781 | 0.4944 |
| MW | 0.0197 | **0.8554** | 0.0030 | 0.2787 |
| BCW | 0.1766 | **0.8479** | −0.2165 | −0.0034 |
| OH | −0.0287 | 0.3039 | 0.0392 | 0.6229 |
| ZW | 0.1389 | 0.3070 | −0.3332 | 0.6959 |
| POC | 0.1169 | 0.1589 | **−0.7789** | 0.0429 |
| RW | 0.2931 | 0.2531 | 0.0622 | 0.6742 |
| RL | 0.4299 | 0.3268 | 0.5776 | −0.1496 |
| CC | 0.4059 | 0.0960 | −0.0271 | **0.7712** |
| MM | 0.3745 | 0.0619 | −0.4280 | 0.6985 |
| CM | **0.7334** | 0.3366 | 0.0206 | 0.4562 |
| PM | **0.7123** | 0.1227 | −0.1282 | 0.4650 |
| C | **0.8001** | −0.1206 | −0.0797 | 0.0967 |
| NO | −0.1033 | 0.5249 | 0.4612 | 0.2852 |
| CM | 0.6273 | 0.2283 | 0.0239 | 0.3402 |
| MDL | 0.4617 | 0.3784 | 0.0390 | 0.6815 |
| MDH | 0.1897 | −0.0661 | 0.0250 | 0.6867 |
| Eigenvalue | 8.0850 | 2.07005 | 1.4810 | 1.1372 |
| % Total | 44.9168 | 11.5003 | 8.2276 | 6.3176 |
| Cumulative variance | 44.9168 | 56.4171 | 64.6447 | 70.9622 |

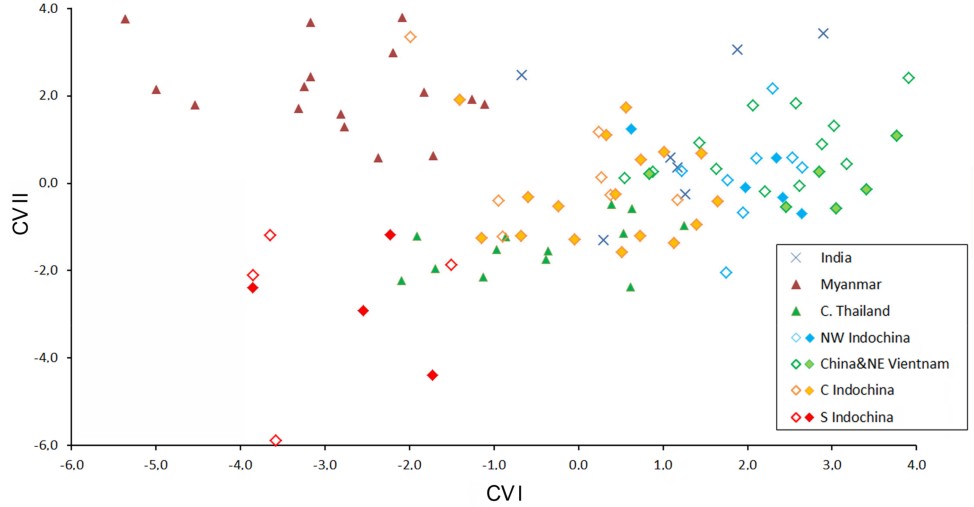

**Figure 6.** Bivariate scatter plot for the first and second canonical variances of a Discriminant Function analysis based on 18 cranial and dental measurements of six training sets of *Hipposideros pomona* s. lato. All Indian specimens were included as "undetermined." Specimens marked by closed diamonds were genotyped with at least one mitochondrial gene.

**Table 5.** Phenetic (squared Mahalanobis) distances between training sets (above diagonal) and significance of difference between them (*p*-level, below diagonal) calculated in a forward stepwise Discriminant Function analysis. Cases of insignificant difference re highlighted in boldface type.

| Training Sets | 1 | 2 | 3 | 4 | 5 | 6 |
|---|---|---|---|---|---|---|
| 1. Myanmar | | 21.3396 | 28.9994 | 31.0698 | 19.3681 | 25.8214 |
| 2. C Thailand | 0.0000 | | 13.5448 | 15.6349 | 10.2886 | 15.3410 |
| 3. NW Indochina | 0.0000 | 0.0000 | | 3.1906 | 7.0932 | 34.5136 |
| 4. NE Indochina/SE China | 0.0000 | 0.0000 | 0.4175 | | 10.6141 | 40.3524 |
| 5. C China | 0.0000 | 0.0002 | 0.0068 | 0.0000 | | 20.9887 |
| 6. S Indochina | 0.0000 | 0.0002 | 0.0000 | 0.0000 | 0,0000 | |

## 4. Discussion

Taxonomical issues regarding the status and delimitation of morphologically similar tropical forms are always quite complicated and cannot be solved by considering one particular trait alone. According to our results, a simultaneous analysis of a few (here seven) nuclear genes can provide a highly supported and reliable tree topology that cannot be obtained from any of the markers separately. The resulting phylogenetic relationships within the *H. gentilis* clade and between this species and other congenerics differs substantially from the previously known mtDNA tree topology. The first remarkable feature of the mitochondrial phylogeny is paraphyletic position of the different *H. gentilis* lineages, which suggests the assignment of species status to at least some of them. However, our nuclear data definitely support the monophyly of *H. gentilis*, which diminishes the likelihood of finding cryptic taxa within this complex. Such discordance may imply historical introgression events.

Specimens from the mountain and lowland regions of southern Vietnam arranged themselves into distinct, but related, haplogroups (also related to the *H. khaokhouayensis/rotalis* lineage) in the mtDNA analysis. However, the nuDNA data is more likely to group animals from different parts of the Vietnamese mountain areas (Dalat plateau, Gialai–Kontum plateaus, and the northern part of central Vietnam) and from northern Vietnam together, suggesting a common origin of these populations. At the same time, the southern lowland branch has full support and is a sister clade to all other *H. gentilis* specimens. Given such topology discordance, we may suppose that the *H. gentilis* populations of southern Vietnam once obtained mtDNA from the common ancestor of the *H. khaokhouayensis/rotalis* lineage (with which they have no direct contact in present). It is difficult to pinpoint the exact time frame of these introgressive events, but they might have happened approximately on the late Pliocene and early Pleistocene boundary (based on the estimated divergence time between *H. gentilis* and *H. khaokhouayensis*: [1]). Noteworthily, the *H. gentilis* population co-occurring with *H. khaokhouayensis* in sympatry on Catba island does not show any traces of such introgression. One may suggest other scenarios leading to the discordance between mitochondrial and nuclear data, for example, ancestral polymorphism, but they seem less likely to us than introgression. The verification of one hypothesis or another requires more material than we currently have, and, probably the employment of additional markers (e.g., microsatellites).

*Hipposideros gentilis* was described from the Thayet (Thayetmyo) district of central Myanmar (Burma) [9,38]. Since it was for long time considered a subspecies of *H. pomona*, all specimens from Myanmar, northeast India, and Thailand were allocated to this form [4,39]. The occurrence of the form *gentilis* in Yunnan and northern Laos was also presumed, mainly from the molecular studies [5].

The form *sinensis* was described from Fujian as a subspecies of *H. gentilis* [4,40]. According to a published study [5] and our original results, this name could be applied to animals from southeastern China and, at least partially, to this species in north Vietnam (e.g., Gulf of Tonkin). Both lineages are quite close to each other, which also agrees with their morphological similarity. It should be noted, however, that the genetically studied material presented in [5] was collected in northeastern Myanmar, far from the terra typica of *H. gentilis*. Specimens which belong to the lines SY and SM (sensu Zhao et al. [5]) are morphologically similar to each other, at the same time the morphologically distinct series from the vicinity of Mandalay demonstrates some difference in skull morphometry (Table 3). The type territory of *H. gentilis* is located even further south (Figure 1). Thus, the interpretation of the SY line as *gentilis* s. str. can be questioned. Perhaps the entire "northern" clade (from south China, north Vietnam, Laos, and the adjoining parts of Myanmar) should be designated as *H. g. sinensis*. The relationship between the later and the nominotypical form requires further clarification.

Animals from central Vietnam (SH clade in [5]) could be regarded as a distinct subspecies, though their distribution limits and relationships with the northern mitochondrial lineages require clarification through more extensive material. The only other named form,

undoubtedly associated with *H. gentilis*, is *megalotis,* which is described from the vicinity of Vinh [11,12]. However, it cannot be used as a valid for this putative subspecies since it represents a junior homonym to the African *Hipposideros megalotis* (Heuglin, 1861).

We can conclude from our data that *Hipposideros gentilis* from the lowland forests of southern Vietnam (and probably from the adjacent areas of Cambodia) is quite a distinct form, both genetically and morphologically. It could be, according to nuclear markers, a sister group for all other *H. gentilis* populations. Meanwhile, no named forms are associated with the "*H. pomona*" (*H. gentilis* in current interpretation) populations from the southern part of the species range [41]. This suggests that the "southern" *H. gentilis* is an undescribed taxon. Its status is not entirely clear, given the common origin and probable past hybridization with other genetic lineages attributed to *H. gentilis*; and it requires further, more comprehensive, study. Therefore, here we refrain from its formal description.

On the whole, we can conclude that *H. gentilis* is, contrary to the previous opinion, a monophyletic taxon, presumably divided into two or three subspecies. Its morphological diversity is to some extent in agreement with the obtained genetic lineages, but the decision regarding the taxonomic rank of these lineages can vary greatly depending on the dataset used (mitochondrial or nuclear markers).

**Supplementary Materials:** The following are available online at https://www.mdpi.com/article/10.3390/d13050218/s1, Table S1: Pairwise distances calculated in cytb gene analysis, Figure S1: Phylogenetic relationships of selected *Hipposideros* species inferred from cytb sequences, Figure S2: Median-joining networks showing the relationships among the alleles of the individual nuclear genes of *Hipposideros gentilis* and some related forms, Table S2: Correspondence between specimen IDs and haplotypes as they designated on Figure S1, Table S3: Correspondence between nodes of the networks on Figure S1 and additional haplotypes, belonging to the same nodes.

**Author Contributions:** Conceptualization, A.P.Y., I.V.A. and S.V.K.; methodology, I.V.A. and S.V.K.; validation, A.P.Y., I.V.A. and S.V.K.; formal analysis, A.P.Y., I.V.A. and S.V.K.; investigation, A.P.Y.; data curation, I.V.A. and S.V.K.; writing—original draft preparation, S.V.K.; writing—review and editing, A.P.Y., I.V.A. and S.V.K.; visualization, A.P.Y., I.V.A. and S.V.K.; supervision, S.V.K.; project administration, I.V.A. and S.V.K.; funding acquisition, S.V.K. All authors have read and agreed to the published version of the manuscript.

**Funding:** Study of collections was supported by the Russian Foundation for Basic Research (grant No. 17-04-00689a). The whole study was performed in line with the stated theme of scientific work of the ZMMU ("Taxonomic and chorological analysis of the animal world, as a ground for study and conservation of the biological diversity", 121032300105-0).

**Institutional Review Board Statement:** Not applicable.

**Informed Consent Statement:** Not applicable.

**Data Availability Statement:** Genetic data: data available in a publicly accessible repository: GenBank and BOLD (see numbers in Table 2 and Appendix B); morphometric data: data available on request from the corresponding author due to privacy reasons.

**Acknowledgments:** We would like to express our thanks to all colleagues who provided their priceless help on different stages of our study. Comparative collection materials were studied by SVK in the Royal Ontario Museum, Canada, due to kind permission of J. L. Eger and B. Lim; in the Museum of Natural History, Great Britain, under the support of R. Portela Miguez; in the Berlin Zoological Museum due to permission and help of Frieder Mayer and Nora Lange; in the Hungarian Natural History Museum, Hungary, under the support of G. Csorba; and in the Natural History Museum of Vienna, Austria, under support of Frank Zachos and Alexander Bibl. Study of the collection materials was done in the Zoological Museum of Moscow University, using the collection facilities, with the support of its director, M.V. Kalyakin. Molecular genetic studies were performed on the facilities of the Vertebrate Zoology Department of Moscow University, with invaluable support from A.A. Bannikova. Obtaining materials from Vietnam became possible through collaboration with the Joint Vietnamese-Russian Tropical Research and Technological Centre, due to the support of Nguyen Dang Hoi and A.N. Kuznetsov.

**Conflicts of Interest:** The authors declare no conflict of interest. The funders had no role in the design of the study; in the collection, analyses, or interpretation of data; in the writing of the manuscript, or in the decision to publish the results.

## Appendix A. List of the GenBank Sequences of the Cytb Mitochondrial Gene, Used in the Analysis

*Hipposideros gentilis*: Laos: DQ054810.1; China (unspecified): KP876550.1; China, Yunnan: DQ888671.1, KJ623705.1, KP336273.1, KP336274.1; China, Guangdong: EU434950.1, KJ619513.1; China, Hainan: KJ623703.1, KJ623704.1; NE Myanmar: MK064112.1, MK064113.1, MK410336.1, MK410337.1, MK410340.1, MK462234.1; Vietnam, Ha Giang: MK091940.1; Vietnam, Quang Tri: MK091943.1; Vietnam, Tuyen Quang: MK091946.1; Vietnam, Phu Tho: MK091947.1, MK091949.1; Vietnam, Ninh Binh: MK410344.1; Vietnam, Binh Thuan: MK410350.1, MK410351.1, MK430028.1; Vietnam, Vinh Phuc: MK430029.1;

*Hipposideros ruber*: EU934477.1, EU934485.1, FJ347985.1, FJ347991.1; *H.* aff. *ruber*: HQ343240.1, HQ343255.1, HQ343258.1; *H. alongensis alongensis*: JN247006.1, JN247007.1; *H. alongensis sungi*: JN247009.1, JN247012.1; *H. armiger*: JN247034.1; *H. armiger terasensis*: JN247045.1; *H. ater*: DQ054807.1; *H. bicolor*: DQ054808.1, MT149741.1, MT149742.1; *H. caffer*: FJ347978.1; *H. calcaratus*: DQ054806.1; *H. cervinus*: DQ054805.1; *H.* cf. *bicolor*: MT149813.1; *H. cineraceus*: DQ054809.1, KX458067.1KX467584.1, LC406452.1, LC406453.1, LC406454.1, LC406456.1, MK091936.1, MK410352.1; *H. diadema*: DQ219421.1; *H. durgadasi*: KY176014.1; *H. fuliginosus*: EU934467.1, EU934468.1; *H. griffini*: JN247040.1, JX849199.1; *H. khaokhouayensis*: DQ054815.1, DQ054816.1; *H. larvatus* s. lato: DQ888672.1, JN247026.1, JN247027.1; *H. lylei*: JN247043.1, KR908662.1; *H. swinhoei*: KJ094477.1, KR908659.1; *H. ridleyi*: DQ054812.1; *H. rotalis*: DQ054813.1, DQ054814.1; *H. sp.* (China): EU434947.1, EU434948.1; *H. pendelburyi*: JN247028.1; *H. turpis*: JN247046.1.

## Appendix B. List of the BOLD Process ID Numbers for Specimens Used in the Analysis

*Hipposideros gentilis*: China, Guangxi: ABCMA599-07, ABCMA601-07, ABCMA609-07; China, Guizhou: ABCMA695-07, ABCMA755-07; Laos: ABBM105-05, ABBM131-05, ABBM210-05, ABRLA158-06, BM160-03, ABBM256-05, ABBM277-05, ABRLA034-06, ABRLA087-06, ABRLA088-06, ABRLA130-06, ABRLA163-06, BM087-03, ABBM336-05, ABBM237-05, ABBM360-05, ABBM375-05, ABRLA066-06, BM036-03; Myanmar: ABBSI116-08, ABBM465-05, BM329-03, BM330-03, BM336-03; Vietnam, Ba Ria-Vung Tau: ABBSI346-11, SKMZM1111-12; Vietnam, Dong Nai: ABBSI259-10, ABBSI404-11, BM618-04; Vietnam, Gia Lai: SKZMR126-19, SKZMR127-19, SKZMR128-19; Vietnam, Hai Phong: SKMZM1097-12, SKMZM1098-12; Vietnam, Kien Giang: BM681-05; Vietnam, Lam Dong: SKMZM1199-13; Vietnam, Quang Nam: ABRVN520-06, ABRVN521-06, ABRVN522-06, ABRVN523-06, ABRVN524-06, ABRVN525-06, ABRVN526-06, ABRVN527-06, ABRVN528-06, ABRVN529-06, ABRVN530-06, ABRVN531-06, ABRVN532-06, ABRVN534-06, ABRVN535-06, ABRVN536-06, ABRVN537-06, ABRVN538-06, ABRVN539-06, ABRVN540-06, ABRVN548-06, ABRVN549-06, ABRVN560-06, ABRVN561-06, ABRVN562-06, ABRVN563-06, ABRVN564-06, ABRVN565-06, ABRVN566-06, ABRVN568-06, ABRVN569-06, ABRVN573-06; Vietnam, Quang Ninh: BM659-05; Vietnam, Tuyen Quang: ABRVN141-06, ABRVN175-06, BM328-04; Vietnam, Vinh Phuc: ABRVN039-06, ABRVN040-06, ABRVN041-06, ABRVN042-06, ABRVN043-06, ABRVN044-06, ABRVN045-06, ABRVN046-06;

*Hipposideros* cf. *bicolor*: Malaysia: BM453-04, BM452-04; Thailand ABBM049-05, ABBM050-05; *H. cineraceus*: Vietnam: ABBSI305-11, ABBSI306-11, ABBSI307-11, ABBSI308-11, ABBSI309-11, ABBSI262-10, ABBSI264-10, ABBSI265-10, ABBSI407-11, BM633-04, BM660-05; *H. galeritus*: Vietnam: ABBSI310-11, ABBSI311-11, ABBSI312-11, ABBSI313-11, ABBSI314-11, SKMZM1126-12, SKMZM1130-12, ABBSI236-10; *H. halophyllus*: Malaysia: ABBSI019-04; *H. khaokhouayensis*: Laos: ABBM376-05, ABBM382-05; Vietnam: SKMZM1108-12; *H. ridleyi*: Malaysia: BM470-04, BM471-04; *H. rotalis*: Laos: BM059-03, ABRLA167-06, ABRLA169-06, ABRLA170-06.

## Appendix C. List of Specimens Used in the Morphometric Analysis. Specimens IDs Genotyped by at Least One Mitochondrial Gene Are in Italic, Specimens Genotyped by Nuclear Genes Are in Bold

India, Haleri: MNH 18.8.3.4 ? (type of *H. pomona*); India, SW India: MNH 2003.397 m; MNH 2003.398 m; MNH 2003.399 m; India, Darjeeling: MNH 21.1.17.78 m; MNH 21.1.17.79 f; MNH 21.1.17.87 m; Myanmar: ZMB 49464 m; ZMB 49465 m; ZMB 49466 m; ZMB 49467 m; ZMB 49471 m; ZMB 49474 m; ZMB 49480 f; ZMB 49482 m; ZMB 49484 m; ZMB 49486 f; ZMB 49487 f; ZMB 49488 f; ZMB 49489 f; ZMB 49491 f; ZMB 49492 f; ZMB 49496 f; China, Guangxi: *ROM MAM 116072 m*; *ROM MAM 116077 m*; China: Guizhou: *ROM MAM 118538 f*; *ROM MAM 118549 f*; China, Fujian: MNH 4.12.2.7 ?; China, Hong Kong: NMNS 06651 m; N. Thailand: ZRC 4.6714 ?; ZRC 4.6715 ?; Thailand, Uthai Thani: NMW 65447 m; NMW 65448 m; NMW 65449 m; NMW 65450 m; NMW 65451 f; NMW 65452 f; NMW 65453 m; NMW 65454 m; NMW 65455 m; NMW 65456 m; NMW 65457 m; NMW 65458 m; NMW 65459 m; Cambodia: HNHM 2005.81.11 f; HNHM 2005.81.13 m; Laos, Khammouan: HNHM 2005.82.46 f; Laos, Tham Phakeo: MNHN 2006-0075 m; MNHN 2006-0076 f; MNHN 2006-0077 m; MNHN 2006-0078 m; Vietnam, Lai Chau: MNH 1997.387 ?; Vietnam, Cao Bang: MNH 1997.326 ?; Vietnam, Tuyen Quang: MNH 1997.383 ?; *ROM MAM 107660 m*; *ROM MAM 107700 m*; Vietnam, Ha Noi: HNHM 88.24.2 f; Vietnam, Cat Ba Island: HNHM 98.90.3 f; ***ZMMU S-190298 f***; ***ZMMU S-190299 m***; Vietnam, Fuong Vong Islands ZMMU S-144297 f; Vietnam, Vinh Phuoc: *ROM MAM 107545 m*; *ROM MAM 107546 f*; *ROM MAM 107547 m*; *ROM MAM 107548 m*; Vietnam, Ninh Binh: HNHM 88.26.1 m; Vietnam, Thanh Hoa: HNHM 88.27.2 m; HNHM 88.28.2 m; HNHM 88.28.3 m; HNHM 88.28.5 m; HNHM 88.29.1 m; HNHM 88.29.2 f; HNHM 88.29.3 f; HNHM 88.29.4 f; Vietnam, Nghe An: MNHN 1947-644 m (type of *"P." megalotis*); Vietnam, Ha Tinh: ZMMU S-164988 m; Vietnam, Quang Binh: *ZMMU S-167172 m*; **ZMMU S-167174 f**; ZMMU S-167175 f; Vietnam, Quang Nam: *ROM MAM 111342 m*; *ROM MAM 111343 f*; *ROM MAM 111344 f*; *ROM MAM 111345 m*; *ROM MAM 111346 m*; *ROM MAM 111347 m*; *ROM MAM 111348 m*; *ROM MAM 111352 f*; *ROM MAM 111353 f*; *ROM MAM 111356 m*; *ROM MAM 111357 m*; *ROM MAM 111371 f*; *ROM MAM 111403 m*; Vietnam, Gia Lai: ***ZMMU S-198153 m***; ***ZMMU S-198154 m***; ***ZMMU S-198155 m***; Vietnam, Dak Lak: ZMMU S-190724 m; Vietnam, Dong Nai: *ZMMU S-175109 f*; **ZMMU S-181871 f**; ZMMU S-186562 m; ZMMU S-194706 m; Vietnam, Phu Quoc Island: ZMMU S-175402 f; *ZMMU S-175403 f*; Vietnam, Ba Ria-Vung Tau: *ZMMU S-188169 m*; ***ZMMU S-190302 m***.

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
