# Peer review of "Not the Cryptic Species: Diversity of Hipposideros gentilis (Chiroptera: Hipposideridae) in Indochina"

_diversity, doi:10.3390/d13050218_

Round 1

Reviewer 1 Report

Dear authors,

thank you for the revision. Please, accept my congrats for the improvement done. Although there are portions of the study I would have done differently, it is your scientific work and it is not the role of the reviewer to force his own way of doing things. I like it and as a bat researcher find the data you presented highly interesting.

Author Response

Thanks for your constructive and friendly input!

Reviewer 2 Report

Overall, I have no major concerns with the content or conclusions of the revised version of the paper. I am generally satisfied with the changes that were made in response to the original reviews.

The grammar is significantly improved, but there are a still quite a few minor grammatical errors, especially in the discussion. It is generally possible to decipher the intention of each sentence, but it would benefit from another round of copy-editing before publishing. For example, on line 13 ‘corroborate’ should be ‘corroborating’, line 29 ‘form’ to ‘from the’. Line 35 ‘single’ should be ‘the single’, etc. There are many more.

There are a few spelling errors including gentilis (line 184, line 289), belongin (line 289) Bayesian (line 197).

A few minor comments:

For Table 2, I suggest for H. ater and H. pomona adding a footnote indicating that names for sequences taken from Genbank follow those used by the original authors. That would clarify why you are still using those names in quotes.

Figure S2 references a Table T1 for the allele names, but I don’t see any table T1. Tables S2 and S3 reference figure S1, but I think they mean S2? Please check all the numbers for figures and tables, as some may have changed but not been corrected.

It would also be useful to explain the meaning of the median-joining network. Not all readers (including myself) will be familiar with these. What are all of the numbers along each of the branches? Are they numbers of substitutions or positions of substitutions?

Author Response

Response to reviewer 2 comments

It is generally possible to decipher the intention of each sentence, but it would benefit from another round of copy-editing before publishing.

Fixed numerous typos and grammar mistakes. Some sentences rewritten for readability.

For Table 2, I suggest for H. ater and H. pomona adding a footnote indicating that names for sequences taken from Genbank follow those used by the original authors.

Footnote added

Figure S2 references a Table T1 for the allele names, but I don’t see any table T1. Tables S2 and S3 reference figure S1, but I think they mean S2? Please check all the numbers for figures and tables, as some may have changed but not been corrected.

Supplementary tables numbering corrected.

It would also be useful to explain the meaning of the median-joining network. Not all readers (including myself) will be familiar with these. What are all of the numbers along each of the branches? Are they numbers of substitutions or positions of substitutions?

Explanation of designations added in caption to supplementary figure 2. Also added supplementary file with matrices of variable characters, used for networks computation.

This manuscript is a resubmission of an earlier submission. The following is a list of the peer review reports and author responses from that submission.

Round 1

Reviewer 1 Report

The manuscript needs to be rewritten. I see the merit very clearly and personally would like to know the news you bring on these bats. Unfortunately, even after repeated study of the text it was not possible to understand the evidence and implications resulting from the data.

The molecular analysis is based on 14 animals from Vietnam. Not a lot, but still it may provide nice and new information. Please, make sure you explain in Introduction, why H. cineraceus is presumed to be part of H. gentilis. It comes as a surprise in Table 1. Perhaps make the Introduction more comprehensible and easy on readers, and in a hierarchical form define the whole small Hipposideros morphogroup and its Asian branch, as delimited by Tate, and land at the target species group. Perhaps another table would be beneficial, with multiple columns referring to various studies and therein suggested organisation of the target bat group.

As to Methods, there is no need to use hipposiderids up to Aselliscus to infer arrangement of 14 sequences of H. gentilis and allies. Please, provide a cytb phylogeny without collapsing clades, with each sequence as a terminal branch. Phylogeny of Hipposideridae is pretty much known, so perhaps the African small Hipposideros might be a satisfactory outgroup. Please, provide genetic distances, they are important for inter- and intraspecific comparison. Also important as support for your discussing of Pliocene/Pleistocene evolutionary events. I also believe that in this case the use of nuclear introns is rather compliant to the current trend in phylogenetic studies than really useful tool for inferring relationships within species. I cannot really tell from your manuscripts what the results mean. MRP may be a tempting modern method of dealing with poorly performing molecular markers, but one has to bear in mind that combining multiple weak signals from slowly mutating nucler introns has to be critically examined before condemning mtDNA. So please, provide networks for each of the introns, and also provide information how alignments were created, and indels and heterozygous sites treated. With regard to this, Table 2 is very reader-unfriendly, consider reworking it. Explain, why it was not possible or necessary to get sequences for each of the newly processed Vietnamese sample, given the low N.

Further, I consider the scheme of assignment of samples and populations to either land or region with clear link between them as rather detrimental to understandability of your results. Please, make clear, what specimens from what country or part of country are linked to the broadly defined regions of eg Indochina etc. It would be best to denote this in the reworked cytb tree, so that the reader is able to connect the sequence with particular morphogroup in the biplots, and moreover to the particular objects within the morphogroups. As to the morphological analysis itself, methodically it is sound, but I would suggest including a simple biplot showing size variation. From the multidimensional analyses it seems to me that the variation among populations/morphogroups is mostly in size. I was wondering what caused the implicit assumption of yours that the morphogroups within the 99 specimens are linked to the phylogroups revealed by the DNA analysis, and subsequently the populations. Please, make it clearer.

I am convinced that after your clarifying of the above mentioned points, the manuscript will be more comprehensible and enable a detailed re-review. Despite my critics, I believe that your study is interesting and the revised manuscript would make a useful contribution to understanding of diversity in the small Asian Hipposideros.

Reviewer 2 Report

Overall, this paper presents a valuable illustration of the importance of considering both nuclear and mitochondrial genetic markers when assessing phylogeny in this group of Hipposideros. The conclusions, including the caveats, about the limitations of using mtDNA to understand phylogenetic relationships in this group are quite important, and appear to be valid. I also agree with the authors’ decision not to describe any new names for any taxa, especially because genetic material in their study was not available from either of the type localities of H. gentilis or H. sinensis.

I do, however, have a few suggestions for ways this paper could be enhanced.

First, there are numerous grammatical errors throughout the manuscript as well as some spelling errors (e.g., Bayesian in Figure 2). I realize that English is not the first language of the authors, but some of these errors make it quite challenging to read and understand some sections, especially in the introduction and the discussion. The paper would benefit form a thorough edit by a native English speaker who also understands the material (to ensure that the correct changes are made).

On Figure 2, it would be helpful to present the sample sizes for each of these nodes – how many different individuals were analysed for each branch – just one or two, or several? It is hard to tell how consistent are the geographic patterns without knowing how many individuals were sampled from each region. The graph gives the impression that these haplotypes are characteristic of each geographic location, but I would not be surprised if some geographic regions have multiple haplotypes unless there is little or no dispersal of females. Larger sampling would help to test that.

If the sample sizes are mostly the same (1 or 2 at each node), perhaps they could be indicated in the figure caption. However, if there is quite a bit of variation in sample size among nodes, perhaps number of specimens could be indicated after each species name or to the right of the tree?

Please explain what is meant by ‘* = maximum support’. Does this mean 100% of replicates?

Line 136-137 suggests this tree largely corroborates a COI tree. As COI data for most or all of these locations are available on Genbank or BOLD (with quite large sample sizes for some), would it be possible to also add a figure using COI? It would be best if the branches were colour coded to match clades/geographic locations from Figure 2 and sorted in the same order (to the extent possible) so the similarities and differences can be readily observed. This would help show whether the geographic patterns are exactly the same, and also help look for situations where multiple haplotypes occur in the same geographic region.

This would also allow inclusion of a number of potentially closely related species that are missing from the cytB graph (such as Hipposideros bicolor and H. kunzi  -- note that most taxonomy on Genbank is likely to call these Hipposideros bicolor or H. atrox or H. bicolor 142 and H. bicolor 132 as the databases have not been updated since the description of H. kunzi, but it should be possible to sort them based on additional information).

Figure 3 – this is the most important figure in the paper, as this is the main novelty of this study. However, there are a few things that could be improved. The authors mention that none of the individual genes provide sufficient resolution for separate analysis, and only in aggregate one can make inferences, although with still insufficient data to estimate distances.

Would it be possible to at least indicate the number of distinct characters actually represented – i.e., how many positions show heterogeneity amongst the samples within the overall H. bicolor / H. gentilis group, or perhaps number of positions that differ between each pair of nodes?

Also, line 174 refers to H. bicolor and H. ater, but these are not actually shown in the tree (assuming the bat labelled “H. ater” from Vietnam is most likely a misidentified H. gentilis). Are there nuclear sequences available for a known H. ater or H. bicolor? If so, why aren’t they on the tree? If not, why are they mentioned in the text? Also, what is the origin of the specimen labelled “H. pomona” on the tree – can it also have the place labelled?

Figures 4 and 5 present some interesting information on morphological variation. These clearly indicate both extensive overlap in morphology among populations, but also some level of geographic variation in average shape. It would seem appropriate to also graph PC3, if it is useful for separating some populations.

The main limitation of these data is that they are not linked, by specimens, to the genetic data. Do any of these morphological specimens correspond to any of the same specimens for which genetic information is available? If so, I strongly recommend labelling the individual dots on these PC graphs for those specimens with numbers that correspond with nodes that were shown on the cytB and nDNA trees. This would help provide some confidence that the morphological patterns actually relate to the DNA patterns.

Overall, the discussion seems reasonable, except that it needs substantial editing for grammar – in some cases I had to guess the intention of a sentence.

Also, the authors talk about introgression as a possible explanation for differences between mtDNA and nDNA. While this is possible, there are other possibilities. For example, if there is very little female dispersal but extensive male dispersal, this could lead to divergence in mtDNA but mixing of nDNA. Under this scenario, is it possible that H. gentilis is an older lineage than H. khaokhouayensis, and the latter evolved from one of the mtDNA branches of H. gentilis? In effect, this could be considered incomplete lineage sorting rather than introgression. It may be worth discussing this as an alternative hypothesis (even if to explain why it might or might not be likely).